# Sparsely Cross-Linked Hydrogel with Starch Fragments as a Multifunctional Soil Conditioner

**Leonid O. Ilyasov** [1,*], **Irina G. Panova** [1], **Petr O. Kushchev** [2], **Andrey A. Belov** [3], **Irina A. Maksimova** [3], **Andrey V. Smagin** [3,4,*] and **Alexander A. Yaroslavov** [1,5]

1 Department of Chemistry, M.V. Lomonosov Moscow State University, Leninskie Gory 1, 119991 Moscow, Russia
2 Department of Chemistry, Voronezh State University, Universitetskaya sq. 1, 394018 Voronezh, Russia
3 Department of Soil Science, M.V. Lomonosov Moscow State University, Leninskie Gory 1, 119991 Moscow, Russia
4 Institute of Forest Science, Russian Academy of Sciences (ILAN), 21 Sovetskaya, Uspenskoe, 143030 Moscow, Russia
5 Eurasian Center for Food Security, M.V. Lomonosov Moscow State University, Leninskie Gory 1, 119991 Moscow, Russia
* Correspondence: illeo98@mail.ru (L.O.I.); smagin@list.ru (A.V.S.); Tel.: +7-903-592-24-04 (L.O.I.)

**Abstract:** A sparsely cross-linked copolymer was synthesized, and was composed of acrylic acid, acrylamide, and starch. Swelling of the copolymer in an aqueous solution resulted in the formation of hydrogel particles; this formulation was used as a partially biodegradable soil conditioner. The hydrogel was characterized with the following main conclusions: (a) the degree of copolymer swelling increases from 300 to 550 when altering the pH of the solution from 3 to 9. (b) After mixing with sand and soil, the degree of swelling decreases because of restricted volumes of sand/soil-filled containers and a mechanical resistance from the sand/soil particles. (c) Initial sand and soil additions demonstrate unsatisfactory water-retaining properties; the addition of the hydrogel significantly increases the maximum water capacity, while a substantial part of the water in the hydrogel remains available to plants. (d) Upon deposition of the hydrogel formulation over sand/soil and drying out, a protective coating forms on the surface, composed of hydrogel and sand/soil particles, resistant to wind and water erosion. (e) The starch-containing hydrogel is non-toxic towards bacterial and fungal microorganisms; the latter can utilize the microgel in order to support their own development. The results of the work indicate that cross-linked anionic copolymers are promising for use as combined soil conditioners.

**Keywords:** cross-linked copolymer; hydrogel; water retention; wind erosion; water erosion; biodegradation; soil conditioner

## 1. Introduction

Polymers with ionic groups, polyelectrolytes (PE), are used in agricultural technologies for the stabilization of soil against wind and water erosion [1–3]. Additionally, PEs are able to preserve/improve the physical properties of arable land and retain moisture and biologically active compounds in the soil [4–6]. When treating the soil with aqueous solutions of linear PEs, protective polymer coatings (crusts) are formed on the surface of soil, which prevents mechanical destruction of the topsoil, erosion of productive soil aggregates, and fine soil particles being blown away [7,8]. Polymeric stabilizers of an amphiphilic nature, for example, ionic polymers with alkyl pendant groups or interpolyelectrolyte complexes prepared from oppositely charged PEs [7], are capable of binding to both the hydrophilic and hydrophobic sites on the surface of soil particles, and have an advantage over traditional PEs [9,10]. However, being effective soil stabilizers, linear polymers bind a limited amount of water and, therefore, have only a slight effect on the water-retention properties of soil [11,12].

In order to improve the water regime of the soil, cross-linked PEs are used with a high content of ionizable functional groups [13–15]. In a water surrounding, such polymers accumulate water whose content is usually many times greater than the weight of the original polymer [16,17]. Water, stored in hydrogels, becomes more available for agriculture and maintains/increases the soil fertility [18,19]; the water pool can carry biologically active compounds [20–22], thus, increasing the efficacy of the agrotechnical hydrogel application. Granular forms of cross-linked polymers composed of acrylamide and acrylic acid salt are generally applied, which, after swelling, are mixed with the root soil layer [23]. Such elastic and mechanically stable hydrogels retain water well, but they cannot pass through small pores and, therefore, cannot be deposited onto the soil surface with a spray machine.

Nowadays, there is a growing interest in soil conditioners capable of degrading when attacked by soil microorganisms and enzymes [23–25]. This approach allows a significant reduction in ecological load on the soil. Here, universal conditioners that combine stabilizing and water-retaining properties with a (bio)degradability and a low toxicity towards soil microorganisms are most in demand.

In the current article, we describe an anionic cross-linked copolymer (ACP#) with biodegradable starch fragments, the swelling of ACP# in an aqueous salt solution, the effect of the resulting hydrogel on the water-retaining properties of quartz sand and a soil with a high sand content (both substrates are widespread in the world's arid and semi-arid regions), and anti-erosion properties of protective polymer–sand and polymer–soil coatings. Additionally, bacteria-induced degradation of starch fragments in the ACP# hydrogel and toxicity of the ACP# hydrogel towards soil microorganisms are presented.

## 2. Materials and Methods

Acrylic acid (AA) ($\geq$98%), starch ($\geq$98%), potassium hydroxide ($\geq$99%), and potassium persulfate ($\geq$98%) were obtained from Vekton (Saint-Petersburg, Russia); acrylamide (AAm) ($\geq$98.5%), a cross-linking agent, N,N-methylene-bis-acrylamide (bisAAm) ($\geq$99%) were acquired from Acros Organics, (Geel, Belgium), KBr ($\geq$98%) and concentrated (35–38 wt.%) HCl ($\geq$98%) from Reakhim (Moscow, Russia) were used as received.

Cross-linked ACP# copolymer was synthesized as follows. A total of 5.5 g of starch was dissolved in 30 mL of distilled water at 80 °C, then 10 mL of 11 wt.% aqueous solution of potassium persulfate was added in order to initiate a graft polymerization of a monomer mixture (see below); the mixture was stirred for 15 min. In parallel, 79.8 mL of 98.5 wt.% aqueous solution of AA and 85 mL of 75.7 wt.% aqueous solution of KOH were mixed at 0 °C, then 20.9 g of Aam and 0.055 g of bisAAm were added. The resulting solution was heated up to 70 °C, then poured into a starch/potassium persulfate binary solution, and the mixture was left under stirring at 60–70 °C until the gel effect appeared. The product was dried in air at room temperature and then in an air flow at 50 °C to constant weight, which showed the quantitative yield of the copolymer, within the accuracy of the gravimetric method ($\pm$0.2%). The dried product was crushed and divided into two fractions with a particle size of <0.25 mm (fraction 1) and 0.25–0.5 mm (fraction 2) by passing it through sieves with a mesh size of 0.25 mm and 0.5 mm. Further experiments were performed with the 0.25–0.5 mm copolymer fraction.

In order to verify the degree of monomer conversion to polymer, 200 mg of the dried product was put in a glass and 200 mL of bi-distilled water was added. Then, 3 days after, the swollen gel was extracted from the glass. The remaining water volume (100 mL) was reduced down to 10 mL using a rotor evaporator and analyzed spectrophotometrically at 200 nm as described in [26]. The total concentration of non-polymerized acrylic/acrylate components in the sample was 3.4 $\mu$g/mL (obtained using corresponding calibration curves) or 0.068 mg in the 200 mL. That corresponded to a 99.966% conversion of the acrylic/acrylate components in the copolymer synthesis.

From there, the maximum content of non-polymerized acrylamide in the copolymer sample did not exceed 0.034%. According to Sojka et al. [27], (co)polymers with a content of



non-polymerized acrylamide of 0.05% and less are considered as non-toxic. This definitely indicates non-toxicity of the synthesized copolymer and related hydrogels.

The composition of ACP# was confirmed by IR spectroscopy using Specord M 80 spectrometer (Carl Zeiss, Oberkochen, Germany). The IR spectrum was recorded in a 3000–1000 cm$^{-1}$ range at reflection regime. Next, 5 mg of ACP# was pressed in a tablet with a dry KBr as matrix. The measurement was carried out at room temperature.

The content of anionic groups in ACP# was quantified via reverse potentiometric titration using an Akvilon pH 420 m (Akvilon, Moscow, Russia). Briefly, a 0.1 wt.% aqueous suspension of APC# was brought to pH ~2–2.5 using a HCl aqueous solution and after 1 h titrated with a 0.1 M NaOH aqueous solution. Concentration of anionic ACP# groups was expressed as the number of moles of anionic groups per liter.

A soil sample was taken in the upper ten centimeter layer of the experimental field near Erokhino village in the Pskov region (Russia) with section coordinates: 56.129474 North and 31.372435 East. The sample was ground and fractionated as described earlier [12]. The granulometric composition (particle size distribution) of the soil sample, obtained by using the laser diffraction method with a Mastersizer 3000E laser particle sizer (Malvern Instruments, Malvern, UK), shows 2.1% for physical clay (a < 0.002 mm fraction), 12.2% for dust (a 0.002–0.05 mm fraction), and the main content of sand (a 0.05–1 mm fraction) with 85.7% (Table 1). According to the USDA classification system, the soil belongs to loamy sandy soil with a low content of organic matter (1.1%), an ECO of 8.8 ± 1.5 mmol (equiv)/100 g, and 0.0057 S/m electrical conductivity of the aqueous extract. The organic matter content was determined by the Tyurin method in Nikitin's modification [28,29]. The conductivity of extracts was measured with an InoLab Cond 7310 Set 1 conductometer (WTW, Weilheim in Oberbayern, Germany); their acidity with an Akvilon 420 pH meter (Akvilon, Moscow, Russia).

**Table 1.** Granulometric composition of the soil sample.

| Size, μm | 0–1 | 1–2 | 2–5 | 5–10 | 10–50 | 50–100 | 100–250 | 250–1000 |
|---|---|---|---|---|---|---|---|---|
| Content, % | 0.7 ± 0.03 | 1.4 ± 0.6 | 2.9 ± 0.14 | 1.8 ± 0.12 | 7.5 ± 0.3 | 2.7 ± 0.14 | 37 ± 2.5 | 46 ± 2 |

Fine-grained monomineralic quartz sand with a grain size of 0.1–0.2 mm (ORT6, Russia) was repeatedly washed with bi-distilled water before use.

An equilibrium degree of free (in air) ACP# swelling (α) was quantified gravimetrically using a Shimadzu MOC-63U moisture analyzer (Shimadzu, Kyoto, Japan). A total of 0.05 g of dry ACP# was placed in 100 mL of bi-distilled water and left for 3 days to swell at room temperature. Then the pH of hydrogel suspension was adjusted to a desirable value with 1M HCl or NaOH aqueous solution, and the suspension was stirred for three days extra to equilibrate. A piece of the swollen hydrogel was placed in the moisture analyzer, weighed, and dried at 100 °C to a constant weight recorded by the device. The α values were calculated according to Equation (1):

$$\alpha = (m_{sw} - m)/m \qquad (1)$$

where $m_{sw}$ and m are the masses of swollen in water and waterless (dry) ACP#, respectively.

In order to quantify swelling of ACP# in a limited pore space of sand and soil substrates (hereinafter referred to as "soil/sand") and to study the effect of ACP# on the water-retention properties of sand and soil, the following procedure was used. The cross-linked copolymer was added to 20 g of sand/soil substrate and mixed, while the substrate/polymer ratio was varied from 100/0.05 to 100/1 *wt/wt*. The resulting mixtures were placed into plastic cups with small holes in the bottom covered with a piece of filter paper; the cups were placed in a container with a 0.001 M phosphate aqueous buffer with a pH of 6.5 and left them 3 days for equilibration. From here, the water content in the

polymer–sand and polymer–soil samples was determined gravimetrically with a Shimadzu MOC-63U moisture analyzer as:

$$W = (M_{sw(mix)} - m_s - m)/M_{(mix)} \tag{2}$$

where $M_{sw(mix)}$ and $M_{(mix)}$ are the masses of wet and dry polymer–sand/soil mixture, respectively, and $m_s$ is the mass of dry sand/soil.

The obtained results were compared with the water content in the original sand/soil at the maximum water saturation, calculated as:

$$W = (m_{sw(s)} - m_{(s)})/m_{(s)} \tag{3}$$

where $m_{sw(s)}$ and $m_{(s)}$ are the masses of wet and dry sand, respectively.

An equilibrium degree of ACP# swelling in a limited pore space of sand/soil ($\alpha_{lim}$) was calculated as:

$$\alpha_{lim} = (M_{sw(mix)} - m_{sw(s)} - m)/m \tag{4}$$

For water retention experiments, 20 mL of a polymer formulation was added to 20 g of sand/soil. The sample was thoroughly mixed for 10 min and dried at room temperature with stirring to constant weight (typically for a week). Water retention curves (WRCs) for the initial sand/soil and ACP#–sand/soil mixtures were obtained by equilibrium centrifugation, as described earlier [12]. The results were plotted in "pF–W" coordinates, where pF is the logarithm of the external pressure created by centrifuge and W is a current water content in the sample.

Of special interest are the points at which the WRCs intersect with secants represented by equations pF = 2.17 + W/100 (5) and pF = 4.18 (6). The intersection of the first with WRC gives the lowest moisture capacity for the sample (field water capacity, FWC), the intersection of the second gives the moisture inaccessible to plants (wilting point, WP) [30]. The difference between these values determines the range of moisture available to plants (available water range): AWR = FWC − WP.

Samples of sand/soil with a protective layer on the surface were prepared as follows. In plastic containers, 60 g of sand/soil were placed, so that the thickness of the layer was 5 cm, the surface area was 16 cm$^2$. Using a spray gun, a 1 wt.% suspension of ACP# hydrogel was applied to the substrate surface with a consumption rate of 2 L/m$^2$, and the samples were dried to constant weight in air. The strength of the polymer–sand/soil crusts was measured by the penetration method using a Rebinder plastometer (Lomonosov MSU, Moscow, Russia) [31].

Anti-erosion experiments were performed following the earlier described procedure [11]. Briefly, a sand/soil sample was placed in a Petri dish, and an ACP# hydrogel suspension was deposited on the top. After drying to constant weight, the samples were tested for the stability to wind erosion by exposing them to an air flow generated by an electric hair dryer for 5–30 min. For the study of water resistance of sand/soil w/wo ACP# [32], the dried samples were treated with 500 mL of water from a spray gun in a pulsed mode for 10 min. After water treatment, the samples were dried, and the weight loss of the sample was calculated. The same was performed with the samples after exposure to air, except for the drying procedure.

Survival of microorganisms (MOs) in the presence of ACP# hydrogel was monitored via cultivation of test cultures on nutrient medium with ACP# as the sole carbon source (1% *w/v*). A mineral base was provided by the Evans medium [33]; MOs by yeasts *Saitozyma podzolica* (VKM Y-1982), *Lipomyces lipofer* (VKPM Y-1877), and *Candida albicans* (VKPM Y-3108); and aerobic bacteria *Bacillus subtilis* (VKM B-501) and *Pseudomonas putida* (VKM B-1301). The biomass of test organisms was suspended in a sterile phosphate buffer saline with pH 7.4, then 5 mL of a liquid medium was inoculated by 100 µL of cell suspension. The samples were incubated at 22 °C for 30 days. The growth of test cultures was monitored visually by altering the turbidity of the samples.

Degradation of the starch fragments in the copolymer, induced by microorganisms, was studied via measuring the viscosity of copolymer hydrogel. For this, 0.11 g of the copolymer was left to swell for a day in 20 mL of a $10^{-3}$ M phosphate buffer with pH 6.5, then the Evans mineral medium was added. Then, prepared solution was inoculated with *Bacillus subtilis* (VKM B-501) to the final cell number of $2 \times 10^2$ CFU/mL. *Bacillus subtilis* strain was selected for this experiment due to its best growth on nutrient medium with ACP# among tested microorganisms and high hydrolytic potential. No thermal sterilization or autoclaving was applied in order to prevent the effect of high temperature and high pressure on physical properties of the hydrogel suspension. The resulting system was kept at 36 °C. The kinematic viscosity was measured immediately after the sample preparation and then every 5–6 days using a VPZh-2 capillary viscometer (Ekrochem, RF) with a capillary diameter of 1.77 mm.

In parallel, every 6–7 days, the number of culturable bacteria in the copolymer–bacterial mixture was quantified as follows. A series of 10-fold sequential dilutions of the mixture in a sterile phosphate buffer with pH 7.4 was prepared. The obtained dilutions were plated on a solid nutrient medium R2A, each dilution was triplicated. The samples were incubated for 14 days, then bacterial colonies were counted using a conventional procedure [34].

## 3. Results and Discussion

### 3.1. Synthesis and Characterization of the Anionic Cross-Linked Copolymer

The ternary copolymer composed of neutral N-acrylamide, anionic potassium acrylate, and starch fragments was synthesized by radical polymerization using the procedure described earlier [35]. The only difference was the use of a stable initiator of polymerization, potassium persulfate, instead of unstable hydrogen peroxide. According to the mechanism of the process [36], the addition of an initiator to a polysaccharide (starch) solution results in the formation of radical sites along the polysaccharide chain and a following graft polymerization of an AA/AAm/cross-linker mixture.

An IR spectrum of ACP# (Figure 1) shows absorption bands, which are characteristic for carboxylic acid salts (1576–1544 cm$^{-1}$ and 1330 cm$^{-1}$) and amides (1672 cm$^{-1}$ and 1400 cm$^{-1}$) [37]. Additionally, there is a wide smoothed superposition of several peaks at the 1125–1000 cm$^{-1}$ range in the spectrum, which is typical for valent vibrations of C-O bonds in carbohydrates [37]. This indicates that starch is part of the copolymer. The IR spectroscopy data, together with a quantitative yield of the copolymer (see Experimental), allows this conclusion about the composition of ACP#: AA/AAm/starch/bisAAm = 14/82/3.96/0.04 weight ratio.

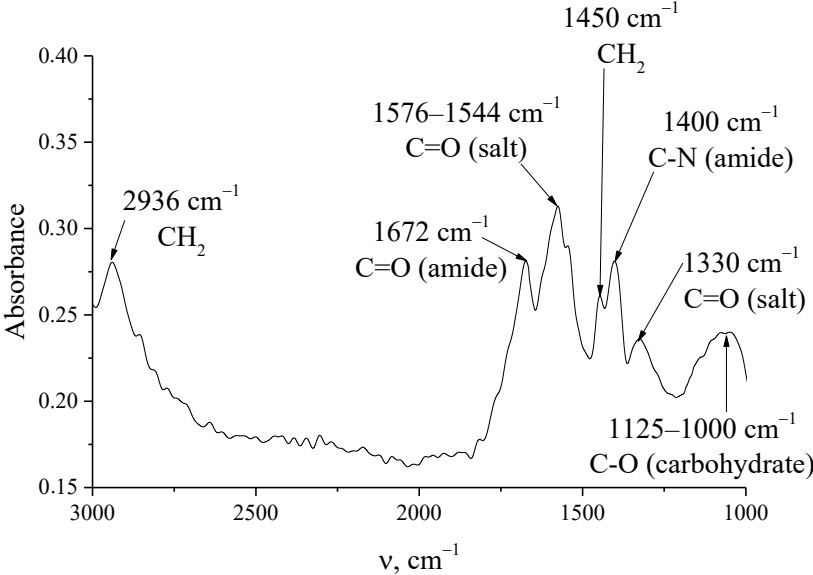

**Figure 1.** Infrared spectrum of ACP#.

The number of carboxylic groups in ACP# was determined with potentiometric titration. First, carboxylic groups were converted to protonated form via addition of 1 M HCl aqueous solution excess followed by titration of the resulting solution with a 0.1 M NaOH aqueous solution (Figure 2). The procedure was repeated three times for statistics (the discrepancy in data was too low to be visible on the graph). The titration curve shows two distinct inflections (leaps) (Figure 2), which correspond to the titration of HCl excess at a $V_1$ point and titration of the ACP# carboxylic groups at a $V_2$ point. These data gave the content of carboxylic groups equal to $6.4 \times 10^{-3}$ moles per 1 g of ACP#.

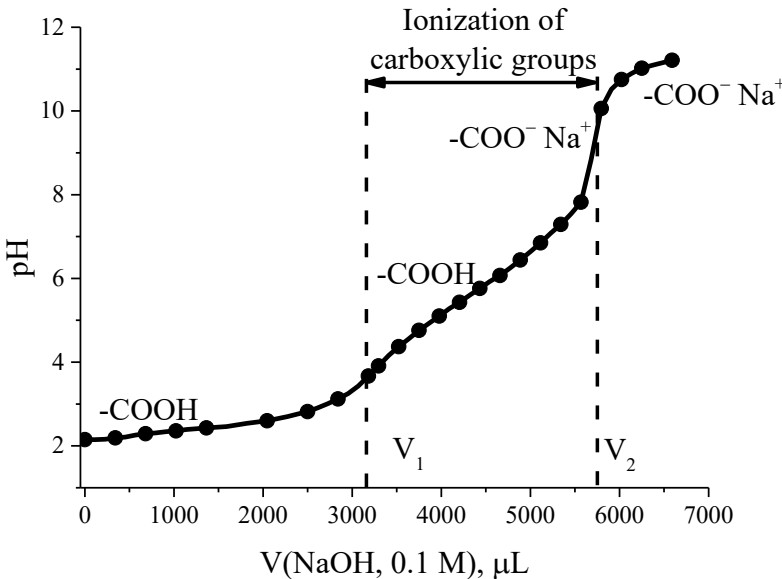

**Figure 2.** Potentiometric titration of 0.1 wt.% ACP# aqueous solution (50 mL) by 0.1 M NaOH aqueous solution. $V_1$ corresponds to titration of HCl excess, $V_2$ to titration of ACP# carboxylic groups (see details in the text).

### 3.2. Swelling of the Cross-Linked Copolymer in Different Environments

In water, dried ACP# particles swell and form transparent hydrogels that spread along the glass surface due to a small degree of cross-linking and the associated low mechanical strength. The degree of ACP# swelling, which is a ratio of the absorbed water weight to the initial dry polymer weight, is sensitive to pH of aqueous solution and increases nearly two times from $\alpha = \sim300$ at pH 3 up to $\alpha = \sim550$ at pH 9. The rise in $\alpha$ is obviously initiated by a progressive dissociation of ACP# carboxylic groups and the unfolding of ACP# hydrogels when increasing pH of aqueous solution.

In reality, the ACP# hydrogels should be mixed with the soil (sand) and, therefore, can be distributed only in a limited pore space of a soil substrate. The volume fractions of pores in a 0.1–0.2 mm quartz sand sample and loamy sandy soil with a granulometric composition shown in Table 1 were quantified earlier [12]. In sand, the size distribution of pores is rather narrow and covers the range from 0.005 to 0.1 mm with a maximum pore size of 0.02 mm [12]. The soil is characterized by a much wider size distribution ranging from 0.0001 to 0.5 mm, with a maximum pore size of 0.03 mm [12]. This fact reflects a wider granulometric composition of soil, in which, among the sand particles, there were species with smaller and larger sizes (see Table 1).

The degree of swelling of ACP# hydrogel after mixing with sand and soil is shown in Figure 3, together with this parameter for ACP# hydrogel swollen in a free state (in air). The swelling occurs in an aqueous buffer with pH 6.5, which is optimum for many agricultural and horticultural crops. The swelling of ACP# in both bulk (loose) substrates dramatically decreases in comparison with the polymer swelling in the free state (in air), due to limited volumes of containers with sand and soil and a resistance from the sand/soil

particles. As a result, the degree of ACP# swelling decreases from $\alpha = 440$ to $\alpha_{lim} = \sim70$ in sand and to $\alpha_{lim} = \sim50$ in soil. Both substrates restrict the ACP# swelling almost equally.

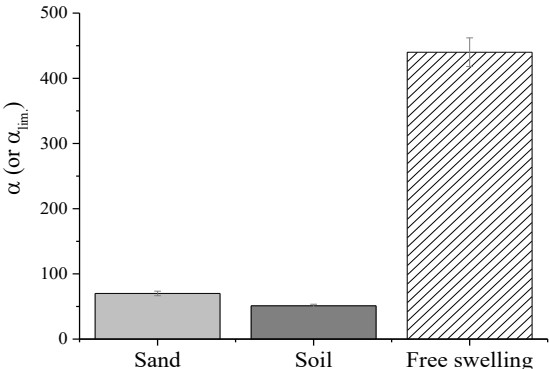

**Figure 3.** The degree of ACP# swelling in the free state and after its mixing with sand and soil. 0.001 M phosphate buffer; pH 6.5.

### 3.3. Effect of the Cross-Linked Copolymer on Water Retention Capacity of Sand and Soil

Water retention capacity of soil substrates is an important characteristic that reflects the accumulation of water available to plants. In our work, the capacity is represented conventionally as a correlation between a weight water content in the substrate, quartz sand, or loamy sandy soil (W), and a logarithm of pressure (pF) specified by sample centrifugation as described elsewhere [12]. The higher the W value at the same pF, the higher the ability of the soil to accumulate and retain water.

Experimentally found W values were approximated by S-shaped curves according to the Van Genuchten model [38]. The maximum water capacity of the initial sand at zero external pressure is equal to $W_{max} = 27\%$ (Figure 4A, curve 1). An increase in pF is, expectedly, accompanied by a decreasing water capacity value, which becomes negligible at pF = 2.5, which corresponds to a pressure of ~31 kPa. The addition of ACP# hydrogel to sand shifts the water retention curve (WRC) to the right, towards a larger water capacity (Figure 4A, curves 2–4). At the same time, the water capacity at an ultimate pF = 4–4.5 also increases and reaches ~30% for a mixture with 1 wt.% ACP# content that even exceeds the W value for the initial sand.

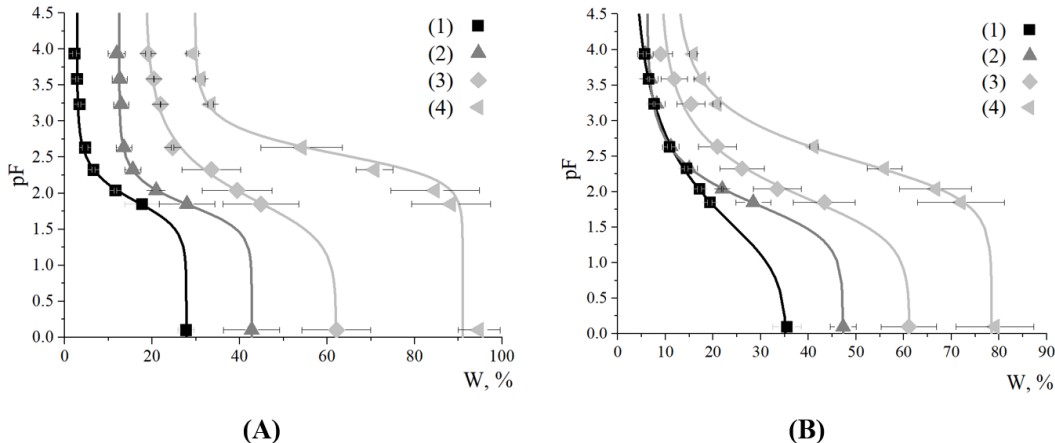

**(A)**          **(B)**

**Figure 4.** Water retention curves for sand (**A**) and soil (**B**) with different contents of ACP#. (**A**) Sand: control (1); 0.2 (2), 0.5 (3), and 1 wt.% of ACP# (4). (**B**) Soil: control (1); 0.2 (2), 0.5 (3), and 1 wt.% of ACP# (4). Samples saturated with 0.001 M phosphate buffer, pH 6.5.

The soil demonstrates a greater maximum capacity for water (Figure 4B, curve 1): 35% against 27% for the initial sand. The addition of ACP# hydrogel increases the $W_{max}$ value

(curves 2–4). However, the water capacity at an ultimate pF = 4–4.5 remains lower than the maximum water capacity of the initial soil even at 1 wt.% ACP#.

The results of WRC analysis for the initial sand/soil and the ACP#–substrate mixtures are summarized in Table 2, in which the maximum water capacities for both substrates at zero external pressure, $W_{max}$, are also shown. In the sand, at a rather high $W_{max}$ = 27%, the FWC does not exceed 4%, and the WP is 1%. The available soil water range is only equal to AWR = FWC − WP = 3 wt.%. In the soil, these parameters are $W_{max}$ = 35%, FWC = 12%, WP = 3, and AWR = 9. The soil shows a greater affinity to water, although, in general, both substrates demonstrate a low water retention capacity.

**Table 2.** Water retention characteristics of sand and soil before and after the introduction of the ACP#.

| Parameter | ACP#, wt.% | | | | | | | |
|---|---|---|---|---|---|---|---|---|
| | Sand | | | | Soil | | | |
| | 0 | 0.2 | 0.5 | 1 | 0 | 0.2 | 0.5 | 1 |
| $W_{max}$ | 27 | 43 | 62 | 91 | 35 | 47 | 61 | 79 |
| FWC | 4 | 16 | 29 | 52 | 12 | 15 | 25 | 43 |
| WP | 1 | 12 | 19 | 30 | 3 | 6 | 10 | 14 |
| AWR = FWC − WP | 3 | 4 | 10 | 21 | 9 | 9 | 15 | 29 |

$W_{max}$, a maximum water capacity. FWC, field water capacity. WP, wilting point. AWR = FWC − WP, available water range.

These indicators improve markedly when the substrates are mixed with the ACP# microgel, including the AWR, the key water-retaining characteristic of soil. The AWR value is reported to be 16% for aggregated fertile loamy soils [39]. Based on this criterion, an ACP# content for an optimum water retention capacity is equal to 0.8 wt.% for sand and 0.5 wt.% for loamy sandy soil.

Summarizing, both substrates, the sand and soil, demonstrate unsatisfactory water-retaining properties. Although the maximum water capacity is 27 wt.% for sand and 35 wt.% for soil, the AWR value, the main hydrophysical indicator, is only 3 and 9 wt.%, respectively. The addition of the ACP# hydrogel leads to a progressive increase in the maximum water capacity of up to 91 wt.% for sand and 79 wt.% for soil at a 1 wt.% content of ACP#. The hydrogel retains the absorbed water after applying a high external pressure: the AWR value is 21 wt.% for sand and 29 wt.% for soil. In other words, a substantial part of water in the ACP# hydrogel remains available to plants.

### 3.4. Stabilization of Sand and Soil with the Cross-Linked Copolymer

The conventional and many times tested procedure for preparing a protective anti-erosion coating is to spray a 1–2 wt.% aqueous polymer formulation over the surface to be treated with a consumption rate of 1–3 L/m$^2$ [11]. However, this simple and effective procedure can only be used for low viscous suspension solutions capable of passing through fine holes of a sprayer. In order to control the viscosity, a 1 wt.% aqueous ACP# suspension was prepared from fraction 2 with <0.25 mm dried particles and tested using a flow viscometer with a capillary of 2.37 mm in diameter. The ACP# suspension easily passes through the viscometer capillary; the kinematic viscosity of the suspension is found to be 32 mm$^2$/s. This suspension could be applied to the sand/soil surface by spraying, which is obviously due to the weak mechanical stability of the ACP# hydrogel particles with a low degree of cross-linking.

Deposition of a 1 wt.% ACP# aqueous formulation over the substrate surfaces leads to formation of polymer–sand/soil coatings that are strong enough to be removed from plastic cups (Figure 5A,B). Typical thickness of coatings is 5–6 mm. The mechanical strength of the coatings was determined by the penetration method using a Rebinder plastometer [31]. The ACP#–sand coating shows a strength of 106 ± 22 MPa. The strength of the ACP#–soil

coating is significantly lower and amounts to 7.6 ± 0.9 mPa. Such a difference obviously results from different composition of both substrates. The sand, consisting only of mineral SiO$_2$ particles, forms a strong composite material (crust) with the ACP# binder, while the soil with 15% of clay, dust, and organic matter forms a less durable composite.

**(A)**                                            **(B)**

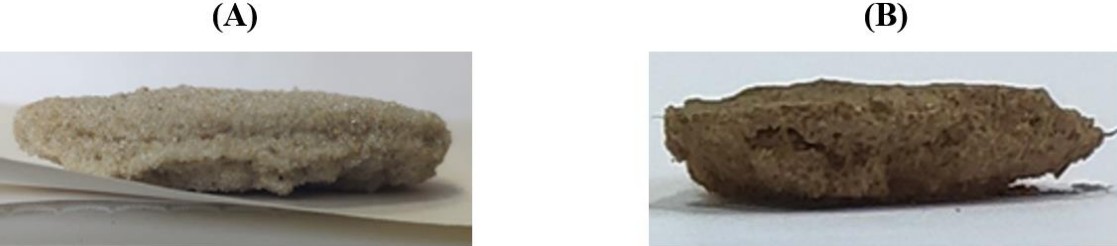

**Figure 5.** Photos of coatings formed on the surface of sand (**A**) and soil (**B**) after their treatment with a 1 wt.% aqueous suspension of ACP#.

The sand and soil, both with and without ACP#, were tested for resistance to an air flow generated by a hair dryer. In control experiments, both sand and soil were completely removed from Petri dishes within a few seconds by an air flow with a speed of 76 km/h. Figure 6 shows the ACP#–sand (A) and ACP#–soil crusts (B) after 30 min exposure to air flow with a 76 km/h speed. In both cases, no removal of substrate is detected, i.e., the ACP#-based crusts demonstrate stability in a storm wind.

**Sand**              **Soil**              **Sand**              **Soil**

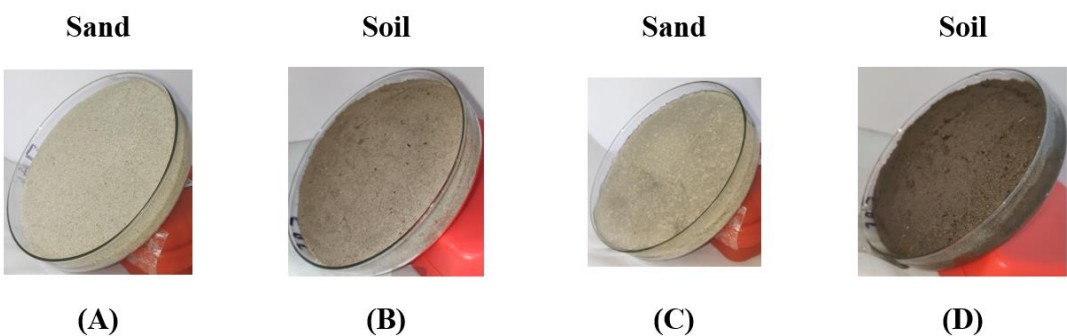

**(A)**              **(B)**              **(C)**              **(D)**

**Figure 6.** Sand (**A**) and soil (**B**) treated with ACP# hydrogel after exposure to wind at a speed of 76 km/h. Sand (**C**) and soil (**D**), both with deposited ACP# hydrogel, after treatment with water.

Additionally, the ACP#–substrate crusts were tested for their resistance to water. First, a control experiment was performed, in which the original sand and soil were subjected to a 10 min treatment with water. The result is 100% wash-out of both substrates from Petri dishes. In contrast to this, the ACP#–sand crust and ACP#–soil crust (Figure 6C,D), when treated with water, quickly swell and turn into jelly-like coatings, which are not washed out by water and completely block soil removal.

### 3.5. Biological Testing of Polycomplex Formulations

A low toxicity towards soil microorganisms (MOs) and biodegradability are important requirements for polymer conditioners [23]. The toxicity was examined with a set of MOs including bacteria widespread in soil: Gram-positive *Bacillus subtilis* and Gram-negative *Pseudomonas putida*, and yeast: *Saitozyma podzolica*, *Limomyces lipofer*, and *Candida albicans*, which are extremely sensitive to toxicants. These cultures are heterotrophic organisms: they are not able to synthesize organic substances for their growth, thus, they use ready-made compounds [40]. The microorganisms were incubated for 30 days at 22 °C in liquid media, which contained ACP# as the only source of carbon and energy, and microelements in order to maintain a vital activity of Mos, but not their growth.

On the first day of the experiment, the samples, composed of ACP#, microelements, and MOs, were nearly transparent. Thirty days after this, all samples turned into very cloudy liquids that definitely showed an active growth of Mos. The results are summarized in Table 3 (row 2), in which the mark "+/+" means survival of MOs and their growth. Thus, the starch-containing ACP# is not only non-toxic towards prokaryotic and eukaryotic microorganisms, but the latter can also utilize the hydrogel as a nutrient medium to support their own development.

**Table 3.** Effect of ACP# on the viability of microorganisms.

| 1 | MO | Bacteria | | | | | | | | | Yeasts | | | | | |
|---|---|---|---|---|---|---|---|---|---|---|---|---|---|---|---|---|
| | | *Bacillus subtilis* | | | *Pseudomonas putida* | | | *Saitozyma podzolica* | | | *Lipomyces lipofer* | | | *Candida albicans* | | |
| 2 | Repeat | 1 | 2 | 3 | 1 | 2 | 3 | 1 | 2 | 3 | 1 | 2 | 3 | 1 | 2 | 3 |
| 3 | Survival/growth | +/+ | +/+ | +/+ | +/+ | +/+ | +/+ | +/+ | +/+ | +/+ | +/+ | +/+ | +/+ | +/+ | +/+ | +/+ |

Finally, biodegradation of starch fragments in ACP# induced by soil microorganisms was examined by monitoring the viscosity of mixed ACP#–MO suspension. It is shown in the control experiment that the addition of MOs, *Bacillus subtilis* bacteria, to an ACP# hydrogel suspension stimulates their progressive growth as represented by *curve 1* in Figure 7: the fast example within the first 12 days and the slower follow-up. Obviously, the bacteria used ACP# starch fragments, which was their only source of food. *Curve 2* in the same figure shows how the viscosity of ACP# suspension alters with time after the addition of *Bacillus subtilis* bacteria. The "viscosity—time" plot consists of three regions. The first, with a progressively decreasing viscosity, reflects a MO-induced cleavage of ACP#. The second, with a slight increase in the viscosity, is probably driven by a release of polysaccharides by bacteria during their vital activity. This increase occurs against the background of the ongoing degradation of starch in ACP#. The third, with a constant viscosity, reflects comparable impacts of ACP# cleavage and MO-induced polysaccharide accumulation. As a whole, the "viscosity—time" plot shows a MO-mediated degradation of ACP# starch fragments.

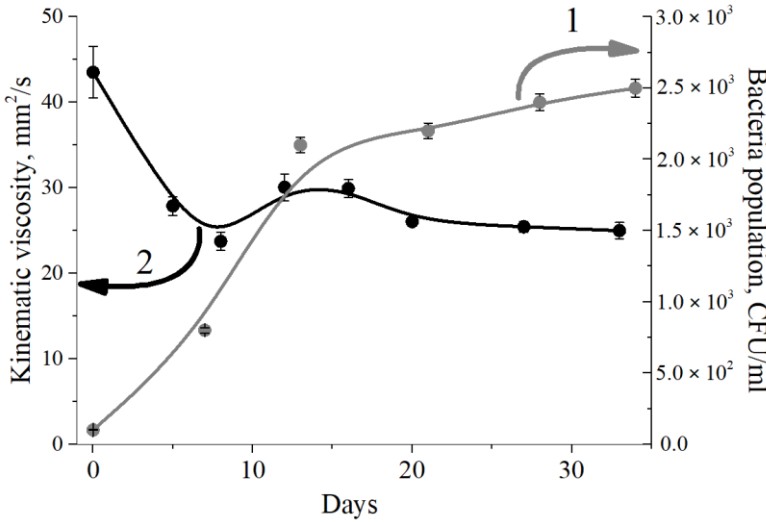

**Figure 7.** The number of *Bacillus subtilis* bacteria in 0.5 wt.% ACP# suspension, prepared in n 0.001 M phosphate buffer (pH 6.5) additionally containing Evans medium (1), and viscosity of the suspension (2) vs. time. 36 °C.

## 4. Conclusions

Synthetic polyelectrolyte networks with a small content of cross-linking agents are of interest as combined soil conditioners. When added to water, such networks swell and

generate hydrogels capable of retaining water and protecting soil against wind and water erosion. The hydrogels spread over the solid surface, adjusting to its relief, thus, forming coatings composed of polymer and soil particles. A rare cross-linking provides an easy mechanical deformation of hydrogel particles and the deposition of an aqueous polymer formulation by spraying that allows a uniform distribution of hydrogel particles in the pores of soil to be treated and a strong binding of individual particles into larger aggregates.

The starch-containing ACP# with a 0.04 wt.% of cross-linking agents, described in the current article, demonstrates the water-retaining effect when only 40–160 mg of the polymer is added to 20 g of sand/soil that corresponds to 0.2–0.8 wt.%. This is comparable with an optimum dose of commercial granular super-absorbents [39]. Additionally, ACP# increases the range of moisture available to plants in sand/soil up to the level of well-aggregated fertile loamy soils [37].

In the aquatic environment, ACP# forms deformable particles so that a 1 wt.% aqueous polymer suspension could be easily applied over the sand/soil surface by using a simple spray gun. It was shown earlier [11] that the efficacy of a linear polymer stabilizers of sand and coarse soil increased with rising molecular mass (MM) of polymers [11]. The transition from a linear polymer to a hydrogel is accompanied by a drastic increase in MM, which could reach several orders of magnitude. Substitution of a linear polyacrylic acid with MM of 100 kDa for an anionic microgel with MM of 80,000 kDa and particles size of 150 nm in the stabilizing formulation leads to a 15-fold increase in the mechanical strength of polymer–soil coatings from 0.22 to 3.3 MPa. The use of millimeter hydrogel particles additionally elevates the coating strength up to 7.6 mPa. When treated with water, the ACP#-based coatings form a jelly-like layer on the sand/soil surface, which prevent a water-induced sand and soil removal.

The starch-containing hydrogel is non-toxic towards bacteria and fungi; moreover, the hydrogel can serve as food for soil microorganisms, thus, supporting their growth and development.

The results of the work indicate that slightly cross-linked anionic copolymers are promising for use as combined soil conditioners.

**Author Contributions:** Conceptualization, I.G.P. and A.A.Y.; methodology, I.G.P., P.O.K., I.A.M. and A.V.S.; software, L.O.I. and A.V.S.; validation, I.G.P. and A.A.Y.; formal analysis, I.G.P. and A.A.Y.; investigation, L.O.I., A.A.B. and I.A.M.; resources, A.V.S., A.A.B., I.A.M. and A.A.Y.; data curation, L.O.I., I.G.P. and A.A.Y.; writing—original draft preparation, I.G.P. and L.O.I.; writing—review and editing, I.G.P. and A.A.Y.; visualization, L.O.I.; supervision, I.G.P. and A.A.Y.; project administration, A.A.Y.; funding acquisition, A.A.Y. All authors have read and agreed to the published version of the manuscript.

**Funding:** This research was funded by the Russian Foundation for Basic Research, grant number 19-29-05036.

**Data Availability Statement:** Not applicable.

**Conflicts of Interest:** The authors declare no conflict of interest. The funders had no role in the design of the study; in the collection, analyses, or interpretation of data; in the writing of the manuscript; or in the decision to publish the results.

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
