# Peer review of "Sparsely Cross-Linked Hydrogel with Starch Fragments as a Multifunctional Soil Conditioner"

_jcs, doi:10.3390/jcs6110347_

Round 1

Reviewer 1 Report

Ilyasov et al present an extensive study on starch-based hydrogels for improving water retention in soil. 

There are few points to address, else conduction of the study and the results are fine. 

- typical abbreviation for acrylamide is AAm and not Am

- concentration of HCl is missing

- no purity for the chemicals is given

- zero temperature should read 0°C 

- wavenumbers have to be changed to 3000 - 1000 cm-1

- the symbol Mw is used for weight average molecular weight and thus ambiguous

- chemical characterization could be extended, especially some spectroscopic confirmation of grafting and a little more information about the calculation of the composition. 

- potentiometric titration is not really conclusive. Has the curve been reproduced? What are the datapoints?

I agree, that the chemical part is probably not the main focus of the study. Adding more information, however, improves the paper. Thus, I recommend acceptance after considering the above mentioned issues.

Author Response

We are grateful to the reviewer for the helpful comments. Here is our reply to the addressed points:

1-1) - typical abbreviation for acrylamide is AAm and not Am

Response: Corrected.

1-2) - concentration of HCl is missing

Response: Corrected.

1-3) - no purity for the chemicals is given

Response: Corrected.

1-4) - zero temperature should read 0°C

Response: Corrected.

1-5) - wavenumbers have to be changed to 3000 - 1000 cm-1

Response: Corrected.

1-6) - the symbol Mw is used for weight average molecular weight and thus ambiguous

Response: Corrected to Msw.

1-7) - chemical characterization could be extended, especially some spectroscopic confirmation of grafting and a little more information about the calculation of the composition.

Response: The article describes the synthesis of a cross-linked copolymer of sodium acrylate and acrylamide, additionally containing biodegradable starch fragments. The yield of the product was found to be quantitative. From there, the ratio of the components in the product is equal to their ratio in the initial reaction mixture. The IR spectrum of the product confirms the presence of all claimed components.

1-8) potentiometric titration is not really conclusive. Has the curve been reproduced? What are the datapoints?

Response: Corrected – data points and information about reproducibility were added.

Reviewer 2 Report

The authors propose the use of acrylic acid-based hydrogels with a net negative charge to increase the water retention of soil and provide protection against environmental factors such as wind and water erosion. The main purpose of the study is to increase the soil quality for agricultural applications. The issue with the manuscript is, however, the use of a synthetic polymer system, acrylic acid and acrylamide with bisAM crosslinker. Acrylamide is a potent cyto-, geno- and neurotoxin, and bisAM could be fatal when ingested. The authors evaluated compatibility of soil/hydrogel samples with microorganisms, but toxicity to animals and humans would be a bigger concern. Therefore, authors should consider using polyelectrolytes of biological origin  (such as polysaccharide alginate) with mild crosslinking approaches for soil treatment rather than synthetic polymer systems with toxic components. 

Author Response

We are grateful to the reviewer for the insight regarding our work. We agree with the reviewer in his/her comment about a high toxicity of acrylamide monomers. In Experimental part of our article, the synthesis of a cross-linked copolymer is described, and a quantitative yield of the copolymer was shown. The latter indicates the absence of free (non-polymerized) acrylamide and bisAM crosslinker in the product. A low toxicity of the copolymer to microorganisms is in a good agreement with an experimentally-found quantitative yield of the copolymer.

Reviewer 3 Report

This manuscript on applying starch-based hydrogels as soil conditioners demonstrates the increase of water capacity of sand and soil by adding hydrogel particles. I would suggest accepting the manuscript after minor revision.

Here are my suggestions for improving the manuscript:

The manuscript needs editing of the English language as well as a spell check. Here are a few examples. There are some more in the text.

Line 34: Polymer*s* with ionic groups …

Line 78: The “acid” in AA acid may have to be deleted according to the authors abbreviation defined earlier in the text.

Line 309: *The* conventional …

Regarding the scientific content: Can the authors please describe the method of determining the swelling degrees using the moisture analyzer in more detail? I think, a reader who is not educated in the field may not be able to intuitively understand the method.

Otherwise, the experiments and discussion thereof are sound.

Author Response

We are grateful to the reviewer for the helpful corrections. Here is our reply to the addressed points:

3-1) The manuscript needs editing of the English language as well as a spell check. Here are a few examples. There are some more in the text.

Response: The manuscript has been corrected in terms of grammar and spelling.

3-2) Line 34: Polymer*s* with ionic groups …

Response: Corrected.

3-3) Line 78: The “acid” in AA acid may have to be deleted according to the authors abbreviation defined earlier in the text.

Response: Corrected.

3-4) Line 309: *The* conventional …

Response: Corrected.

3-5) Regarding the scientific content: Can the authors please describe the method of determining the swelling degrees using the moisture analyzer in more detail? I think, a reader who is not educated in the field may not be able to intuitively understand the method.

Response: Additional explanations were provided in text.

Round 2

Reviewer 1 Report

with the additions, the manuscript can be published

Author Response

Dear and highly respected Reviewer #1! Thank you very much for your positive assessment of the manuscript and permission to publish it in J. Compos. Sci.. We have corrected the text, eliminating typos and grammatical errors in the revised version of the manuscript.

From the team of authors with best wishes, Prof. Andrey V. Smagin

Reviewer 2 Report

Authors addressed in their comment the concerns of potential toxicity with no residual crosslinker and no toxicity on microorganisms. However, it is not clear in the experimental section how the quantification of individual components was performed. The only method used to characterize chemical composition of the gel was IR spectroscopy, which is not sufficient to quantify the presence of unreacted crosslinker and initiator. Acrylamides are potent neurotoxins for mammals (it can even be absorbed through unbroken skin even if not swallowed), which could not be addressed by tests on prokaryotes. Thus, the non-toxicity of the hydrogels on microorganisms does not indicate their safety regarding higher organisms in wild life, farmers, or human consumption of the produce grown in these soil samples. As stated in the first revision, a polymer of biological origin with non-toxic crosslinker components is highly recommended to avoid concerns over soil quality and safety regarding wild life and human consumption.  

Author Response

Dear and highly respected Reviewer #2! Thank you very much for your positive assessment of the manuscript .  In the new version of the manuscript, we tried to take into account your comments as much as possible. Below we comment on the main points.

General consideration before discussion of details. Acrylamide-based polymer constructs are widely used in agriculture as soil conditioners, in particular, for anti-erosion soil stabilization and water retaining. They are commercial large-tonnage products. But these polymers are not (bio)degradable and require long time to break down.

We suggest to render biodegradability to the conventional polymers and polymeric hydrogels via incorporation a native polysaccharide (starch) in their structure. We show how polysaccharide fragments can be incorporated in the cross-linked polymers, provide physico-chemical characteristics of the resulting hydrogels and describe the cytotoxicity of hydrogels to soil microorganisms (MOs). The latter is a key point since a high toxicity to MOs is blocking their way to agriculture. For this reason, we began with MOs and we are planning to use the more complicated test-objects at the next stages of work.

Emphasize, we describe the principal methodology for the synthesis of biodegradable hydrogels but not a protocol for fabricating a commercial product for agricultural application. From the above it should be quite clear why we do not discuss in the article the non-toxicity of the hydrogels “regarding higher organisms in wild life, farmers, or human consumption of the produce grown in these soil samples”.

1) Nevertheless, we used an additional procedure for detecting a possible presence of potentially toxic acrylamide and a cross-linker, N,N-methylene-bis-acrylamide, in the final product, cross-linked ACP copolymer.

In order to verify the degree of monomer conversion to polymer, 200 mg of the dried product was put in a glass and 200 mL of bi-distilled water was added. 3 days after the swollen gel was extracted from the glass. The remaining water volume (100 mL) was reduced down to 10 mL using a rotor evaporator and analyzed spectrophotometrically at 200 nm as described in [26]. The total concentration of non-polymerized acrylic/acrylate components in the sample was 3.4 µg/mL (obtained using corresponding calibration curves) or 0.068 mg in the 200 mL. That corresponded to a 99,966% conversion of the acrylic/acrylate components in the copolymer synthesis.

From there, the maximum content of non-polymerized acrylamide in the copolymer sample did not exceed 0.034%. According to Sojka et al. [27], (co)polymers with a content of non-polymerized acrylamide of 0.05% and less are considered as non-toxic. This definitely indicates non-toxicity of the synthesized copolymer and related hydrogels.

2) A vanishingly small content of free acrylamide in the product is in accordance with the low toxicity of the product, which was proved by the microbiological results.

We realize that MOs are not the only object to quantify the toxicity of hydrogels. It is only the first step to receive an entire toxicity picture. However, it is a key point, which opens a long procedure for testing toxicity of new “agriculture-oriented” products. Now we know that the hydrogels are not toxic to soil MOs and can come to other test-objects.

3) As to recommendation of the reviewer about a polymer of biological origin for the hydrogel synthesis, we will definitely use it when planning our next steps with polymer hydrogels.

I hope we were able to satisfy you on the main points of the review.

From the team of authors with best wishes, Prof. Andrey V. Smagin

Reviewer 3 Report

After revision of the manuscript by the authors, I would suggest accepting in the present form.

Author Response

Dear and highly respected Reviewer #3! Thank you very much for your positive assessment of the manuscript and your permission to publish it in the present form.

From the team of authors with best wishes, Prof. Andrey V. Smagin